# Effects of two substrates at different phosphorus levels on morphology and physiology of *Dianthus barbatus* Linn

Gui Chen[1☯], Zhihuan Mao[1☯], Danhong Yin[1]*, Ling Liu[1], Xin Bin[1], Yike Qin[1], Shengping Zhao[1], Aoqin Ma[1], Tengfei Huang[2]

1 College of Agriculture and Forestry Ecology, Shaoyang University, Shaoyang, Hunan, China, 2 Shaoyang Forestry Bureau, Shaoyang, Hunan, China

☯ These authors contributed equally to this work.
* yindanhong0810@163.com

**Data Availability Statement:** All relevant data are within the paper and its Supporting Information files.

## Abstract

*Dianthus barbatus* linn. is widely used in gardens, mainly as flower beds and flower borders. The effects of different gradients of P on the growth and root morphology of *Dianthus barbatus* were studied to explore its morphological and physiological responses and adaptive strategies. Hence, this study provides a theoretical basis and practical guidance for *D. barbatus* production. Two soil substrates, namely loess and vegetable soil, and five phosphorus concentration gradients were set; no phosphorus application was used as the control. The morphology and physiology of *D. barbatus* were also investigated. Low-to-medium- and low-phosphorus treatments promoted the growth of *D. barbatus* in the above and underground parts of the plants grown on both substrates. Chlorophyll content, flower quantity, and acid phosphatase activity in the rhizosphere soil were significantly increased in the $H_1$ and $H_2$ treatments of loess and in the $C_4$ treatment of vegetable soil. Thus, *D. barbatus* seems to reduce the damage caused by phosphorus stress by increasing chlorophyll content and root acid phosphatase activity. The latter was significantly higher in vegetable soil than in loess. Vegetable soil was more conducive to *D. barbatus* growth than loess.

## Introduction

In the plant body, phosphorus (P) one of is the main nutrient element for plant growth. It is an important constituent of organic compounds such as nucleotides and proteins, participates in many physiological and biochemical reactions in the plant body, and plays a crucial role in plant development [1] The main cultivated soil in southern China is loess. Owing to its unique physical and chemical properties, elemental P in this type of substrate is easily fixed [2]. Vegetable soils are usually obtained from the topsoil of vegetable gardens or leguminous crops, which have good fertility and agglomerate structure, while loess soil is slightly acidic and relatively poor in nutrients [3]. However, fixed P cannot be directly absorbed by plants, and the P applied as fertilizer is generally wasted [4]. Some studies have shown that the utilization rate of P in loess is only 10–20% [5]. Therefore, the physiological mechanisms of P uptake and

**Funding:** This work was sponsored in part by Natural Science Foundation of Hunan Province (2022JJ50228) and National innovation and Entrepreneurship training program for college students(202110547016).

**Competing interests:** The authors have declared that no competing interests exist.

utilization in plants and the physiological response of plants to P deficiency stress must be thoroughly understood to find the most suitable P concentration for plant growth and to improve the efficiency of P uptake and utilization in plants.

Root traits play an important role in plant growth and reflect plant adaptations to soil nutrients' availability [6]. In a study on root morphology and nutrient accumulation in plants, Kang et al. found that P application significantly increased the total root length, specific root length, and root surface area. Additionally, the accumulation of P in the root system and whole plant increased, indicating that root length and root surface area were the main strategies for plants to respond to changes in water and fertilizer availability [7]. Acid phosphatase is an important hydrolytic enzyme in both plants and soil; it can hydrolyze P fixed by ions and increase the soil available P content [8], playing an important role in the decomposition and utilization of organic P in soil [9]. Furthermore, the activity of acid phosphatase in the rhizosphere is significantly correlated with the content of organic P in the soil [10]. The content of available phosphorus in the rhizosphere soil was significantly higher than that in the non-rhizosphere soil [11]. Xu Jing et al. showed that the activity of acid phosphatase in the rhizosphere soil of wild barley under low P stress increased significantly and was significantly higher than that under high P treatment [12].

The growth of plants is inseparable from photosynthesis, and chlorophyll content affects the speed of photosynthesis [13]. *Dianthus barbatus*, belonging to family Caryophyllaceae, is a perennial herb commonly known as confetti carnation, sweet William, Chinese pink, Ten Kinds of Brocade, or American carnation. Although *D. barbatus* originated in Europe, it is cultivated throughout China and widely used in landscaping because of its bright and rich color, long flowering period, and strong adaptability [14]. This small plant has many flowers and cymes, flowers in spring and summer, and generally has annual and biennial flowers. It can be used in flowerbeds, as a potted plant, in rock gardens, or as a lawn-edge ornament. In large cultivation areas, *D. barbatus* can be used as landscape ground cover or cut flower material. In terms of ecology, *D. barbatus* absorbs sulfur dioxide and chlorine gas and can be used for plant greening [15]. There are many studies on the effects of P addition on plant roots worldwide but there has been no research on how P affects the roots of *D. barbatus*. Therefore, the present study aimed to provide practical value for systematically exploring the effects of P addition on the above- and underground parts and physiological characteristics of *D. barbatus*.

## Materials and methods

The present study was carried out in the greenhouse of Shaoyang University at an altitude of 230 m. The area is rainy and hot and south winds blow in the summer while northern winds blow in winter. The experiments began in September 2021; *D. barbatus* plants were harvested in June 2022, replanted (one plant per pot), and culled according to the emergence and growth of the seedlings within half a month of the beginning of the experiment.

Two different substrates were set, namely loess and vegetable soil. According to the soil agrochemical analysis method [16], the loess soil has the following components: Soil organic carbon was 16.2 g/kg, soil total phosphorus was 0.47 g/kg, soil total potassium was 11.8 g/kg, total nitrogen was 1.67 g/kg, and pH was 5.42, the vegetable soil has the following components: soil total phosphorus was 0.85 g/kg, soil total potassium was 16.8 g/kg, total nitrogen was 1.80 g/kg, and pH was 6.10, and five different gradients of P were applied (Table 1). Phosphorous pentoxide ($P_2O_5$), which was administered as the P source, was mixed into the soil matrix according to the experimental design. After thorough mixing, 5 kg of soil were transferred into each pot, loaded into a nutrient bowl (diameter, 30 cm; height, 28 cm). Robust seedlings were planted in each nutrient bowl with six replicates per gradient. The growth of *D. barbatus* as well as the morphological and physiological characteristics of its root system were analyzed.

**Table 1. Experimental treatment and fertilization.** Loess(H). Vegetable soil(C).

| Treatment | $P_2O_5$/(g $Kg^{-1}$) | Treatment | $P_2O_5$/(g $Kg^{-1}$) |
|---|---|---|---|
| $H_0$ | 0 | $C_0$ | 0 |
| $H_1$ | 0.192 | $C_1$ | 0.192 |
| $H_2$ | 0.384 | $C_2$ | 0.384 |
| $H_3$ | 0.576 | $C_3$ | 0.576 |
| $H_4$ | 0.768 | $C_4$ | 0.768 |

## Determination of morphological indexes

**Height and fresh weight of whole plants.** The height and fresh weight of whole plants were randomly measured in six plants from each P treatment group. Plant height was measured from the top to the lowest end of each plant and a mean heigh was determined. An electronic balance was used to quantify the fresh weight of the whole plant, as well as that of the above- and underground parts.

**Root biomass.** The length of each root was measured with a ruler and the length of the main and fine roots was obtained for each sample; these lengths were then averaged to obtain the total root length. Ten random root lengths were selected in each P gradient. The upper and lower ends of the roots were measured with Vernier calipers to calculate the average root diameter. The drainage method was used to measure the volume of the main and fine roots. The water stains were wiped off from the surface of roots and the fresh weight of roots was determined in an electronic balance. The roots were then placed in a clean envelope and dried at 70°C for 48h before determining the root dry weight in an electronic balance.

**Leaf biomass.** The harvested *D. barbatus* was cleaned with distilled water, and 18 fresh leaves were flattened, washed wi deionized water, and wiped to remove the excess water. After determining the leaves fresh weight using an electronic balance, they were spread on a scanner to obtain leaf images. The leaf image files were then imported into Photoshop, and the leaf area was calculated. A Vernier caliper was used to calculate the thickness of 18 leaves. The leaves were placed in a plastic reagent bottle and soaked for 24 h, then removed to wipe the surface water stains, and finally the saturated fresh weight of the 18 leaves was measured. Leaves and other plant parts were placed in a clean envelope and dried at 70°C for 48 h. The dry weight of the leave, as well as that of the aboveground parts was determined using an electronic balance.

**Flower quantity.** Counting the number of buds.

**Chlorophyll concentration.** Leaf blades were cut into small pieces (0.05 g; length of 1 mm) and transferred into a 15 mL test tube($A_1$) containing 75% of 5mL. The test tube was placed into an electromagnetic wave cleaner for 10 min and 3.5 mL of the solution from the test tube($A_1$) was transferred into a cuvette for spectrophotometry analysis at 633, 645, and 663 nm wavelengths.

**Conductivity.** Leaf blades were cut into small segments (0.05 g; length of 3 mm) and placed into a test tube containing 10 mL of deionized water. The test tube was placed into a magnetic stirrer, stirred for 30 min, and the conductivity was measured. The tube was placed into a hot bath (100°C), and the conductivity was measured again.

**Soil acid phosphatase activity.** The sodium diphenyl phosphate colorimetric method was used [17]. The phosphatase in this experiment was measured using the "Soil acid phosphatase (S-ACP) activity detection kit". The measuring principle of the kit is that S-ACP catalyzes the hydrolysis of disodium phenyl phosphate to phenol and disodium hydrogen phosphate in an acidic environment, and S-ACP activity can be calculated by measuring the amount of phenol

produced. The kit contains: reagent one (liquid 21ml), reagent two (powder), reagent three (liquid 11ml), reagent four (powder), standard product (liquid 1ml), all reagents are stored at 4˚C.

Visible spectrophotometer preheat more than 30min, adjust the wavelength to 660nm, and zero the distilled water. Blank tube: Take 1mL glass colorimetric dish, add 50μL of reagent one, 200μL of reagent three,20μL of reagent four, mix thoroughly, add 730μL of distilled water after color development, after mixing, stand at room temperature for 30min, measure absorbance at 660nm, and record it as $A_{blank\ tube}$.

Standard tube: Take 1mL glass colorimetric dish, add 50μL of standard liquid, 200μL of reagent three, 20μL of reagent four, mix thoroughly, add 730μL of distilled water after color development, mix and stand at room temperature for 30min, measure absorbance at 660nm, and record as $A_{standard\ tube}$

Measurement tube: Take 1ml glass colorimetric dish, add 50μL of supernatant, 200μL of reagent three, 20μL of reagent four, mix well, add 730μL of distilled water after color development, mix and stand at room temperature for 30min, measure absorbance at 660nm, and record as $A_{measurement\ tube}$

$ACP(U/g\ soil\ sample) = [C_{standard\ liquid}x(A_{measurement\ tube} - A_{blank\ tube}) \div (A_{standard\ tube} - A_{blank\ tube})]$ $xV_{total}x1000 \div W \div T = 725x(A_{measurement\ tube} - A_{blank\ tube}) \div (A_{standard\ tube} - A_{blank\ tube}) \div W$

$C_{standard\ liquid}$: 0.5μmol/ml; $V_{total}$: Total volume of catalytic system, 1.45ml; W: Soil sample quality, g; T: Catalytic reaction time, 24h = 1d;

## Data statistics and analysis

Morphological indicators, such as specific root length, root specific surface area, root tissue density, and root-crown ratio were calculated according to the following formula:

Specific root length (m/g) = fine root length / fine root dry weight;

Specific surface area ($cm^2$/g) = fine root surface area / fine root dry weight;

Root tissue density (g/$cm^3$) = dry weight of fine roots / volume of fine roots;

Root-to-shoot ratio (R/S) = dry weight of underground part / dry weight of aboveground part.

Specific leaf weight, specific leaf area, leaf dry matter content, leaf tissue density, leaf relative water content and other morphological indicators are calculated according to the following formula:

Specific leaf weight (g/$cm^2$) = leaf stem mass / leaf area;

Specific leaf area ($cm^2$/g) = leaf area / leaf stem mass;

Leaf dry matter content (g/g) = leaf stem quality / leaf saturated fresh mass;

Leaf tissue density (g/$cm^3$) = leaf stem mass / (leaf area x leaf thickness);

Relative water content of leaves (%) = (fresh leaf quality—leaf stem mass) / (leaf saturated fresh mass—leaf stem mass) x 100%.

Chlorophyll content (mg·dm$^{-2}$) was calculated by the modified Arnon formula [18].

Chlorophyll a content = $12.7A_{663}$-$2.59A_{645}$;

Chlorophyll b content = $22.88A_{645}$-$4.67A_{633}$;

Total chlorophyll content = $20.29A_{645}$+$8.05A_{633}$;

In the formula, $A_{633}$, $A_{645}$, and $A_{633}$ absorb at wavelengths of 633, 645, and 663 nm, respectively. Used Excel2019 for the original data processing in the early stage, and related statistical analysis such as one-way ANOVA analysis, normality test, independent sample T test, and one-way ANOVA analysis were carried out through SPSS19.0, and use Origin to make the histogram (S1 Table).

## Results and analysis

### Plant height and flower quantity

Plant height intuitively reflects the growth status of plants. It was determined under different levels of P supply for the loess substrate. The height of *D. barbatus* in the $H_1$ treatment was the highest, which was significantly higher than that in the other treatments (Fig 1A). The effects of $H_2$ treatments on plant height were slight but significantly higher than that of medium-, high-, and no-P treatments (Fig 1A). The flower quantity in the $H_1$, $H_2$ treatments was significantly higher than that in the $H_3$, $H_4$, and $H_0$ treatments (Fig 1A). The plant height and flower number of *D. barbatus* first increased and then decreased with increasing P concentration in the loess substrate (Fig 1A). Therefore, lower P concentrations in loess substrates can promote plant growth and flower numbers.

When the substrate was vegetable soil with different levels of P supply, *D. barbatus* height was significantly higher in the $C_1$ and $C_4$ treatments than in the other treatments (Fig 1B). The flower quantity in the $C_4$ treatment was the highest, which was significantly higher than that in the other treatments (Fig 1B).

When the substrate was loess soil, the growth of *D. barbatus* was poor in the $H_4$ treatment (Fig 1), as plants were short and flowers were few. When the substrate was vegetable soil, *D. barbatus* grew well under the $C_4$ treatment. Plant height and flower number were significantly higher than in the other treatments. In both substrates, *D. barbatus* presented a good condition under the low-P treatment. Under the higher levels of P supply, *D. barbatus* plants in vegetable soil were generally taller and had more flower (Fig 1). Thus, vegetable soil was helpful for the vegetative and reproductive growth of this species.

### Leaf morphological index

Under different levels of P supply, when the substrate was loess, the leaf area was largest under the $H_2$ treatments, which was significantly higher than that under other treatments (Fig 2A). The leaf area under the $H_4$ treatment was significantly higher than that under the $H_0$ and $H_3$ treatments (Fig 2A). The leaf area under the $H_1$ treatment was significantly higher than that

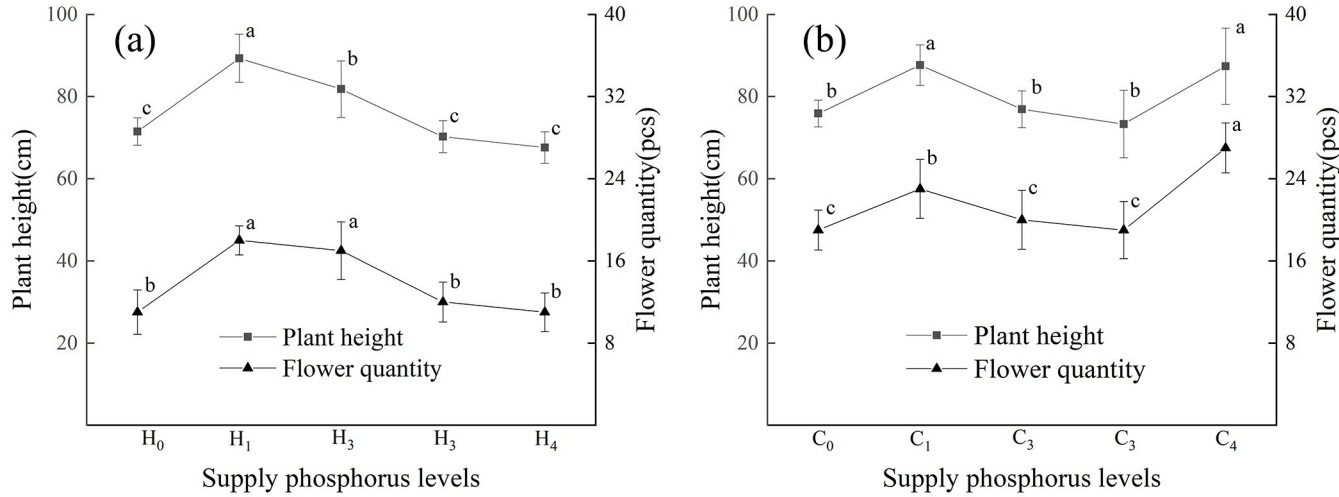

**Fig 1. Effects of different phosphorus levels on plant height and flowering number of *Dianthus barbatus* Linn.** Different lowercase letters indicate significant differences among different phosphorus treatments under the same substrate at the 0.05 level, The same below. Plant height (a), flowering quantity (b) under different P levels. (The diagram used one-way ANOVA analysis).

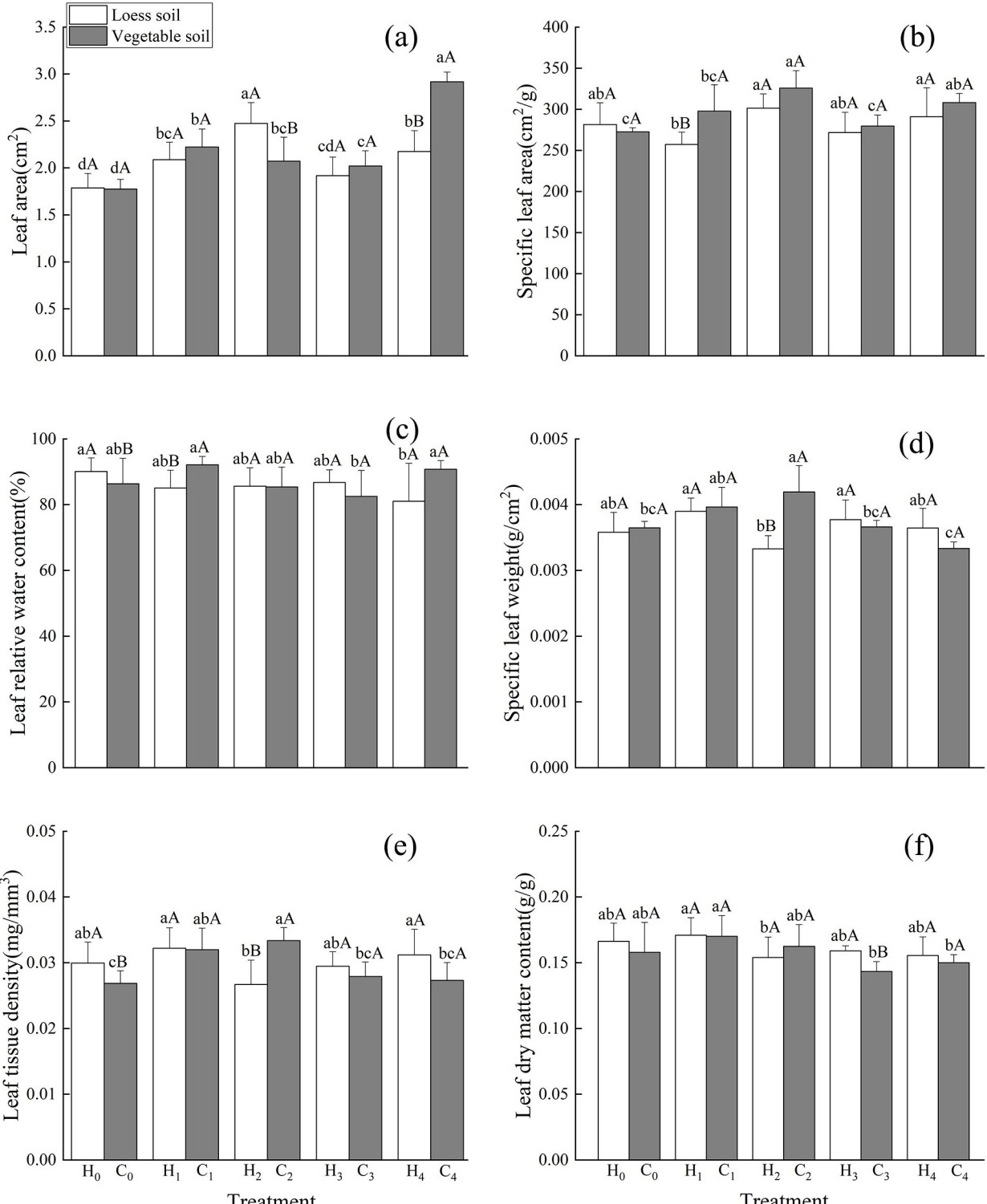

**Fig 2. Effects of different phosphorus levels on leaf morphological indices of *Dianthus barbatus* Linn.** Different capital letters indicate significant differences between two substrates with the same phosphorus treatment at the 0.05 level. Leaf area (a), specific leaf area (b), leaf relative water content (c), specific leaf weight (d), Leaf tissue density (e)and leaf dry matter content (f) under different P levels. (The diagram used one-way ANOVA analysis).

under the $H_0$ treatment (Fig 2A). The specific leaf area was largest under the $H_2$ treatments, which was significantly higher than that under the $H_0$, $H_1$, and $H_3$ treatments (Fig 2B). The specific leaf area under the $H_4$ treatment was significantly higher than that under the $H_0$ and $H_3$ treatments (Fig 2B). The leaf relative water content under the $H_0$ treatment was the highest, which was significantly higher than that under the $H_4$ treatment (Fig 2C). The specific leaf weight under the $H_1$ treatment was the highest, which was significantly higher than that under the $H_2$ treatments (Fig 2D). The specific leaf weights under the $H_2$ treatments were significantly lower than those under the other treatments (Fig 2D). The leaf tissue density was the highest under the $H_1$ treatment, followed by the $H_4$ treatment, which was significantly higher than that under the $H_2$ treatments (Fig 2E). The leaf dry matter content was the highest under the $H_1$ treatment, which was significantly higher than that under the $H_2$ treatments (Fig 2F).

Under different levels of P supply and in vegetable substrate, leaf area was the largest under the $C_4$ treatment, which was significantly higher than that under the other treatments (Fig 2A). The leaf area of the $C_1$ treatment was significantly higher than that of the $C_3$ and $C_0$ treatments (Fig 2A). The leaf area under $C_0$ treatment was the lowest and decreased by 64.39% compared to that under the $C_4$ treatment (Fig 2A). The specific leaf area was largest under the $C_2$ treatment and significantly higher than that under the $C_0$, $C_1$, and $C_3$ treatments (Fig 2B). The specific leaf area under the $C_4$ treatment was significantly higher than that under the $C_0$ and $C_3$ treatments (Fig 2B). The leaf relative water content under the $C_1$ treatment was the highest, followed by the $C_4$ treatment, both of which were significantly higher than that under the $C_3$ treatment (Fig 2C). The specific leaf weight under the $C_2$ treatment was the highest, which was significantly higher than that under the $C_0$, $C_3$, and $C_4$ treatments (Fig 2D). The specific leaf weight under the $C_1$ treatment was significantly higher than that under the $C_4$ treatment (Fig 2D). The specific leaf weight under the $C_1$ treatment was significantly higher than that under the $C_4$ treatment (Fig 2D). The leaf tissue density under $C_2$ treatment was the highest, which was significantly higher than that under the $C_0$, $C_3$, and $C_4$ (Fig 2E). The leaf tissue density under the $C_1$ treatment was significantly higher than that under the $C_0$ treatment (Fig 2E). The leaf dry matter content under the $C_1$ treatment was the highest, which was significantly higher than that under the medium- and $C_4$ treatments (Fig 2F).

Compared with the loess substrate, the leaf relative water content and leaf tissue density decreased significantly in the vegetable soil without P treatment (Fig 2C and 2E). Under the low-P treatment, the specific leaf area and relative leaf water content increased significantly (Fig 2B and 2C). Under the medium- and low-P treatments, the leaf area decreased significantly, while the specific leaf weight and leaf tissue density increased significantly (Fig 2A and 2D and 2E). Under the medium-P treatment, leaf dry matter decreased significantly (Fig 2F). Leaf area was significantly increased under the high-P treatment (Fig 2A).

## Root morphological index

Under different P levels, the root morphology of *D. barbatus* changed (Fig 3A). When the substrate was loess, the total root length was highest in the $H_2$ treatment, followed by the $H_3$ treatment. Both treatments had significantly higher values than the other treatments (Fig 3A). The effect of $H_1$ treatment on total root length was not significant compared to that of $H_0$ treatment (Fig 3A). The total root length under the $H_4$ treatment was significantly reduced compared to that under the $H_0$ treatment (Fig 3A). The root volume of the $H_2$ and $H_3$ treatments was significantly higher than that of the $H_0$ and $H_4$ treatments (Fig 3B). The specific root length of low phosphorus treatment was the largest, which was significantly higher than other treatments (Fig 3C). The fine root surface area under the $H_1$ and $H_2$ treatments was significantly higher than that in the $H_4$ treatments (Fig 3D). There was no significant difference between the

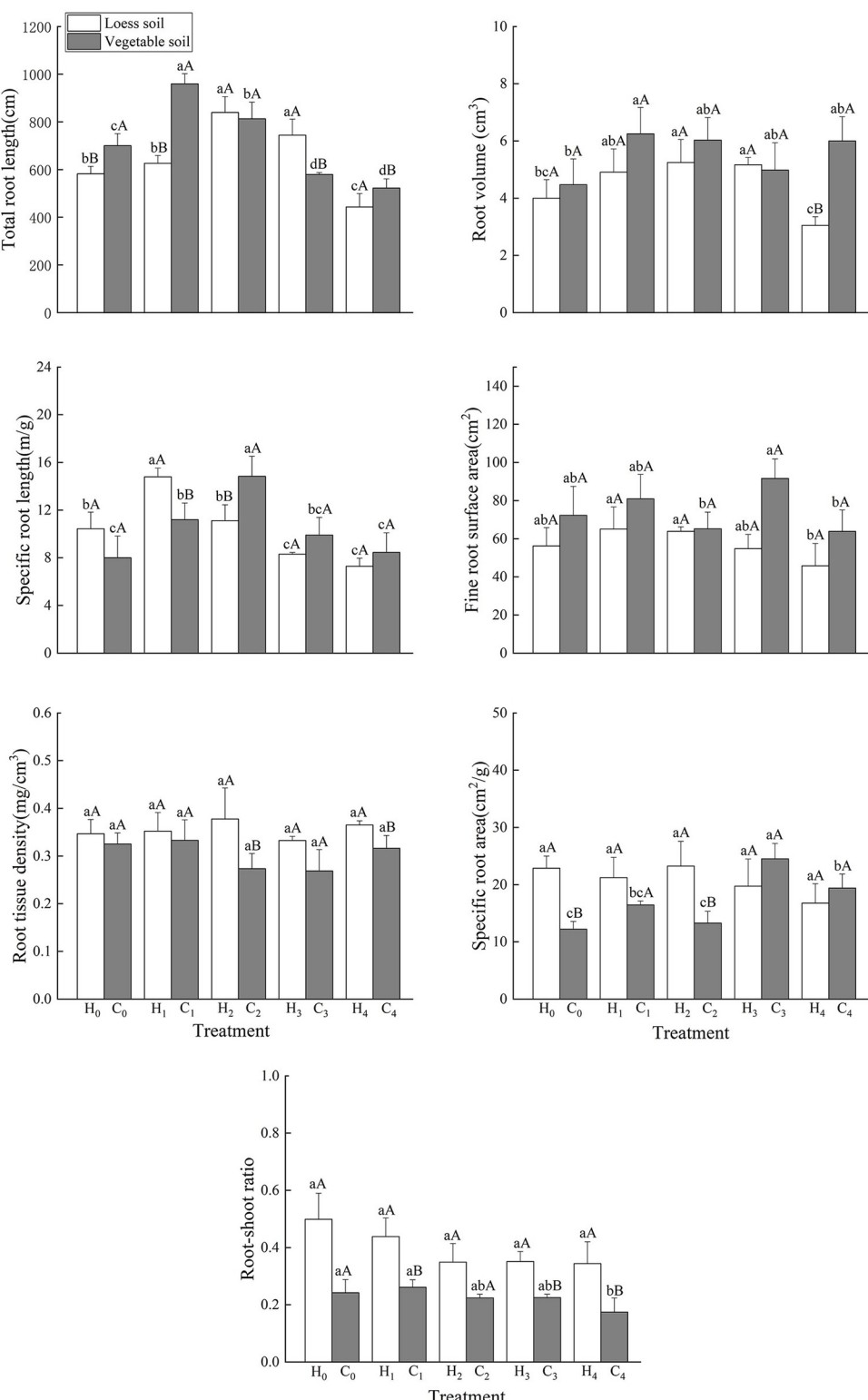

**Fig 3. Effects of different phosphorus levels on subterranean parts of *Dianthus barbatus* Linn.** Total root length (a), root volume (b), specific root length (c), fine root surface area (d), Leaf tissue density (e), root tissue density (f) and root-shoot ratio (g) under different P levels. (The diagram used one-way ANOVA analysis).

specific surface area and root tissue density at the different P concentrations (Fig 3E and 3F). The root/shoot ratio was the highest in the $H_0$ treatment and decreased with increasing P concentration (Fig 3G).

When the substrate was vegetable soil, the total root length under the $C_1$ treatment was the largest and significantly higher than that under the other treatments (Fig 3A). The total root length under the $C_3$ and $C_4$ treatments was significantly reduced compared to that under the $C_0$ treatment (Fig 3A). The root volume reached a maximum in the $C_1$ treatment, which was significantly higher than that in the $C_0$ treatment (Fig 3B). The Specific root length under the $C_2$ treatment was the largest and significantly higher than that under the other treatments (Fig 3C). The second highest value was found under the $C_1$ treatment, which was significantly higher than the $C_0$ and $C_4$ treatments (Fig 3C). The fine root surface area was the largest under the $C_3$ treatment, which was significantly higher than that under the $C_2$ and $C_4$ treatments (Fig 3D). The root tissue density was the highest under the $C_1$ treatment, but there was no significant difference in the root tissue density between different P concentrations (Fig 3E), and the specific root area under the $C_3$ treatment was the largest, which was significantly higher than that under the other treatments(Fig 3F). The root/shoot ratio was the highest under $C_1$ treatment, followed by $C_0$ treatment, and significantly higher in both treatments than that under the $C_4$ treatment (Fig 3G). The root/shoot ratio decreased with increasing P concentrations.

Compared with the loess substrate, the total root length increased significantly, and the specific surface area decreased significantly in the vegetable soil without P treatment (Fig 3A and 3F). In the vegetable soil, and under low-P conditions, the total root length increased significantly while the specific root length and root/shoot ratio decreased significantly (Fig 3A and 3C and 3G). Under the middle-to-low-P treatment, the specific root length increased significantly, the specific surface area and root tissue density decreased significantly, and the root/shoot ratio also decreased, but there was no significant difference (Fig 3C and 3E and 3F and 3G). The total root length and root/shoot ratio significantly decreased under medium-P conditions (Fig 3A and 3G). Under the high-P treatment, the total root length and root volume significantly increased, and the root tissue density and root/shoot ratio significantly decreased (Fig 3A and 3B and 3E and 3g).

## Chlorophyll

Chlorophyll content affects the photosynthetic capacity of the leaves. Under the different levels of P supply, when the substrate was loess, an increase in P concentration led to an initial increase in chlorophyll a, chlorophyll b, and total chlorophyll content, followed by a decrease (Table 2). Chlorophyll a was the highest in the $H_2$ treatment, which was significantly higher than that in the $H_0$,$H_3$, and $H_4$ treatments (Table 2). The $H_1$ treatment showed the second highest value, which was significantly higher than that of the $H_0$,$H_3$ treatments(Table 2). The total chlorophyll content was the highest in the $H_2$ treatment, followed by the $H_1$ treatment (Table 2). Both were significantly higher than those in the no-, medium-, and high-P treatments (Table 2). The results indicated that the $H_2$ treatments could help chloroplasts absorb more light energy and produce more photosynthates (Table 2). Chlorophyll a/b was highest in the $H_4$ treatment, which was significantly higher than that in the $H_1$ treatment (Table 2).

Under the different levels of P supply, when the substrate was vegetable soil, the chlorophyll a, chlorophyll b, and total chlorophyll content showed an increasing trend (Table 2). Chlorophyll a and b were the highest in the $C_4$ treatment, and were significantly higher than those in the $C_0$,$C_1$, and $C_2$ treatments (Table 2). Chlorophyll a/b was the highest in the $C_1$ treatment, which was significantly higher than that in the $C_3$ and $C_4$ treatments (Table 2). The total

**Table 2. Effects of different phosphorus levels on chlorophyll content in leaves of *Dianthus barbatus* Linn.** (The diagram used one-way ANOVA analysis).

| substrate | Treatment | chlorophyll a | chlorophyll b | Total chlorophyll content | Chlorophyll a/b |
|---|---|---|---|---|---|
| loess soil | $H_0$ | 5.02c | 3.51±0.64b | 4.69±0.82b | 1.43±0.05ab |
| | $H_1$ | 7.25ab | 5.13±0.9a | 6.83±1.19a | 1.42±0.05b |
| | $H_2$ | 7.85a | 5.38±1.03a | 7.01±1.33a | 1.46±0.02ab |
| | $H_3$ | 2.60d | 1.96±0.35c | 2.69±0.47c | 1.32±0.04c |
| | $H_4$ | 5.76bc | 3.83±0.19b | 5.01±0.23b | 1.5±0.05a |
| vegetable soil | $C_0$ | 5.55c | 3.59±0.36b | 6.04±1.79b | 1.54±0.01ab |
| | $C_1$ | 7.00bc | 3.9±0.6a | 5.19±0.81a | 1.59±0.06b |
| | $C_2$ | 6.05bc | 4.12±0.68a | 5.47±0.87a | 1.57±0.02ab |
| | $C_3$ | 8.13ab | 5.37±1.41c | 7.05±1.91c | 1.52±0.03c |
| | $C_4$ | 8.77a | 6.99±1.83b | 7.52±1.27b | 1.5±0.01a |

chlorophyll content was highest in the $C_4$ treatment, but there was no significant difference among the five P treatments (Table 2).

## Acid phosphatase activity and leaf blade relative conductivity

Whether in bulk soil or rhizosphere soil, the acid phosphatase activity of loess was significantly lower than that of vegetable soil (Fig 4A and 4B). Acid phosphatase activity in the bulk soil of the loess substrate increased with increasing P concentration and was the highest under the $H_4$ treatment, which was significantly higher than that under the $H_0$, $H_1$, $H_2$, and $H_3$ treatments (Fig 4A). Under different levels of P supply, when the substrate was loess, the activity of acid phosphatase in the rhizosphere soil first increased and then decreased (Fig 4B). It was the highest under the $H_1$ treatment, which was significantly higher than that under the other treatments (Fig 4B). The treatment showed the second highest value, which was significantly higher than that under the $H_0$, $H_3$ and $H_4$ treatments (Fig 4B). Under different levels of P supply, when the substrate was vegetable soil, the activity of acid phosphatase in the bulk soil first increased and then decreased (Fig 4A). The $C_3$ treatmentst showed the highest value, which was significantly higher than that under the other treatments (Fig 4A). The $C_2$ treatments showed the second largest value, which was significantly higher than that under the $C_0$, $C_1$ treatments (Fig 4A).

When the substrate was vegetable soil, the activity of acid phosphatase in the rhizosphere soil was the highest under the $C_0$ treatment, followed by the $C_2$ treatment. Both treatments showed significantly higher values than those under the $C_3$ and $C_4$ treatments (Fig 4B). With the increase in soil P concentration, acid phosphatase activity of bulk soil also increased, whereas the acid phosphatase activity of rhizosphere soil was stronger at lower P concentrations.

The leaf blade relative conductivity exhibited different trends for the two substrates (Fig 4C). When the substrate was loess, the maximum value was observed under the $H_1$ treatment, which was significantly higher than that under $H_2$ conditions (Fig 4C). When the substrate was vegetable soil, the maximum value was observed under the $C_4$ treatment, which was significantly higher than that under $C_1$ conditions (Fig 4C). The results showed that *D. barbatus* was affected differently by P stress at the same P concentration in both substrates (Fig 4C).

## Relationship between leaf physiological indexes and aboveground parts

When the substrate was loess, blade relative conductivity had a negative correlation with flower quantity and leaf area and was positively correlated with plant height (Fig 5A and

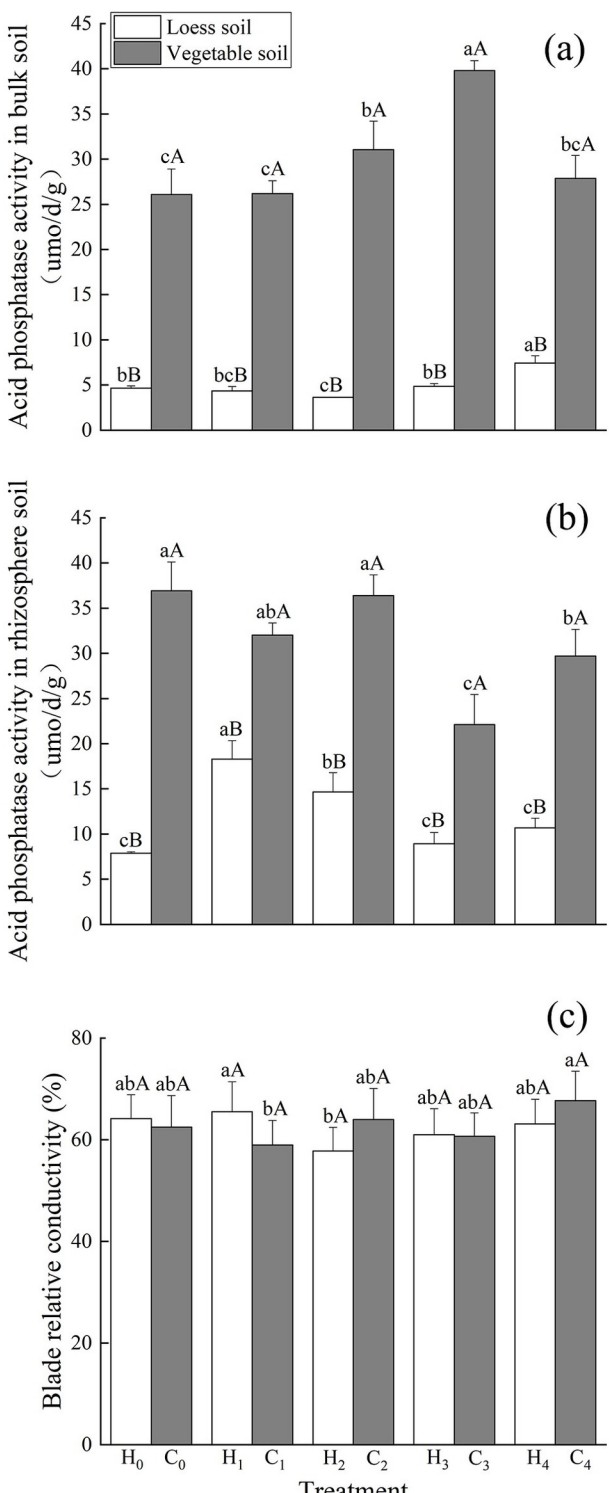

**Fig 4. Effects of different phosphorus levels on acid phosphatase activity and relative electrical conductivity of Dianthus barbatus Linn.** Acid phosphatase activity in bulk soil (a), acid phosphatase activity in rhizosphere soil and blade relative conductivity (c) under different P levels.

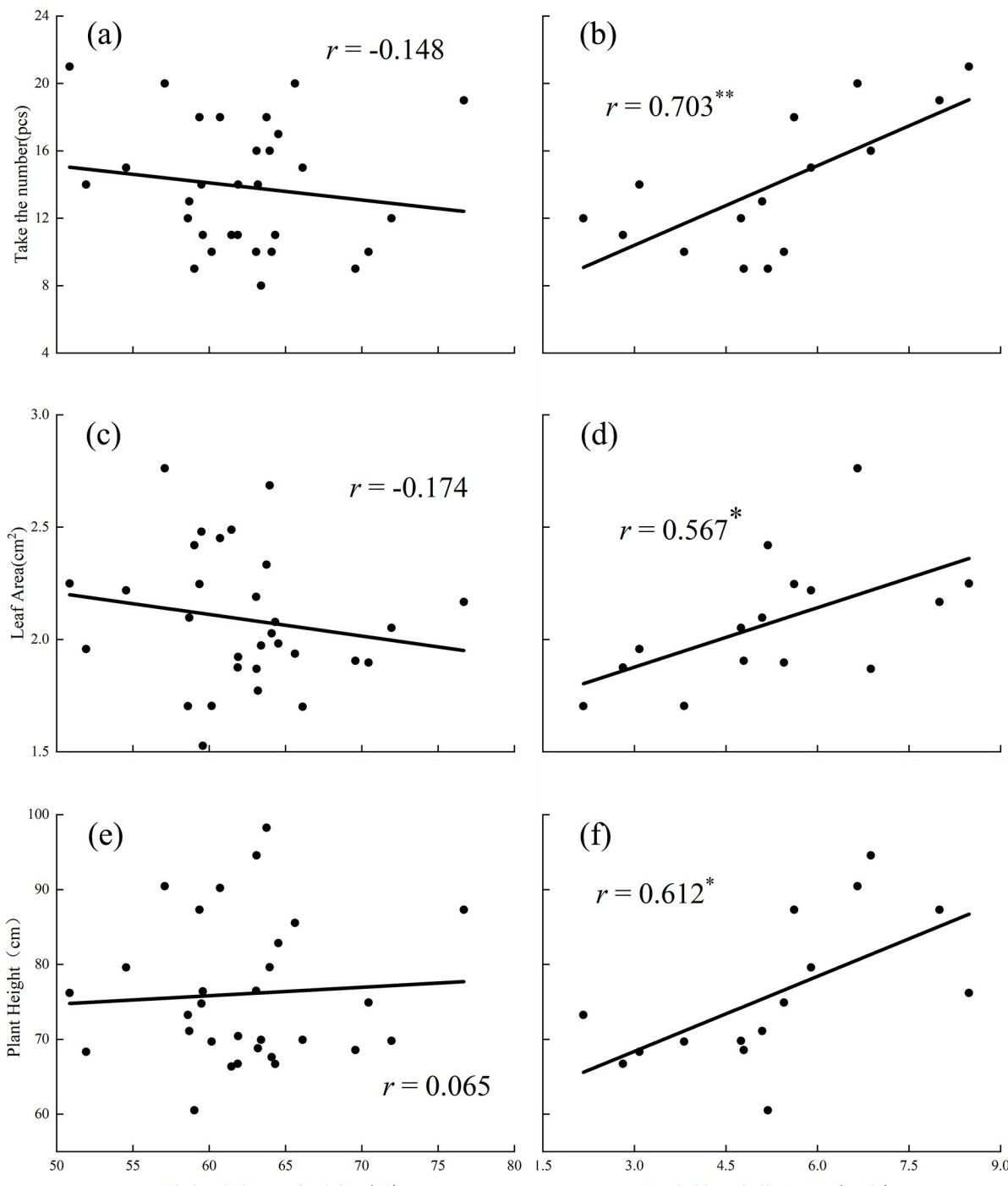

**Fig 5.** Relationships between the blade relative conductivity and flower quantity (a), leaf area (c) and plant height (e) in loess soil. Relationships between the total chlorophyll content and flower quantity (b), leaf area (d) and plant height (f) in loess soil. (The graph uses the statistical method of correlation analysis).

5C and 5E). Total chlorophyll content had a significant positive correlation with plant height and leaf area and was strongly positively correlated with flower quantity (Fig 5B and 5D and 5F).

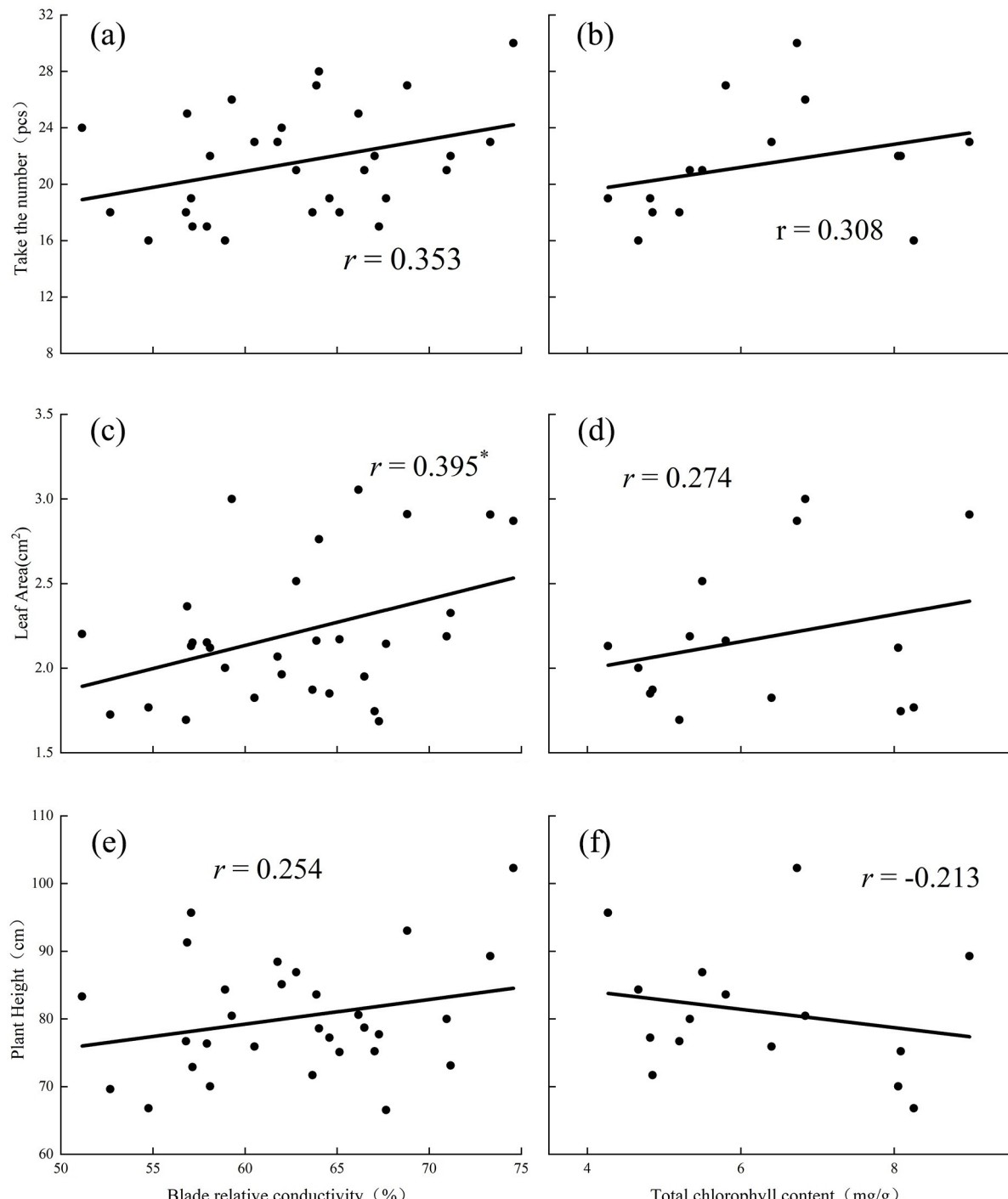

**Fig 6.** Relationships between the blade relative conductivity and flower quantity (a), leaf area (c) and plant height(e) in vegetable soil. Relationships between the total chlorophyll content and flower quantity (b), leaf area(d) and plant height (f) in vegetable soil. (The graph uses the statistical method of correlation analysis).

When the substrate was vegetable soil, blade relative conductivity had a significant positive correlation with leaf area and had a positive correlation with plant height and flower quantity (Fig 6A and 6C and 6E), while total chlorophyll content was positively correlated

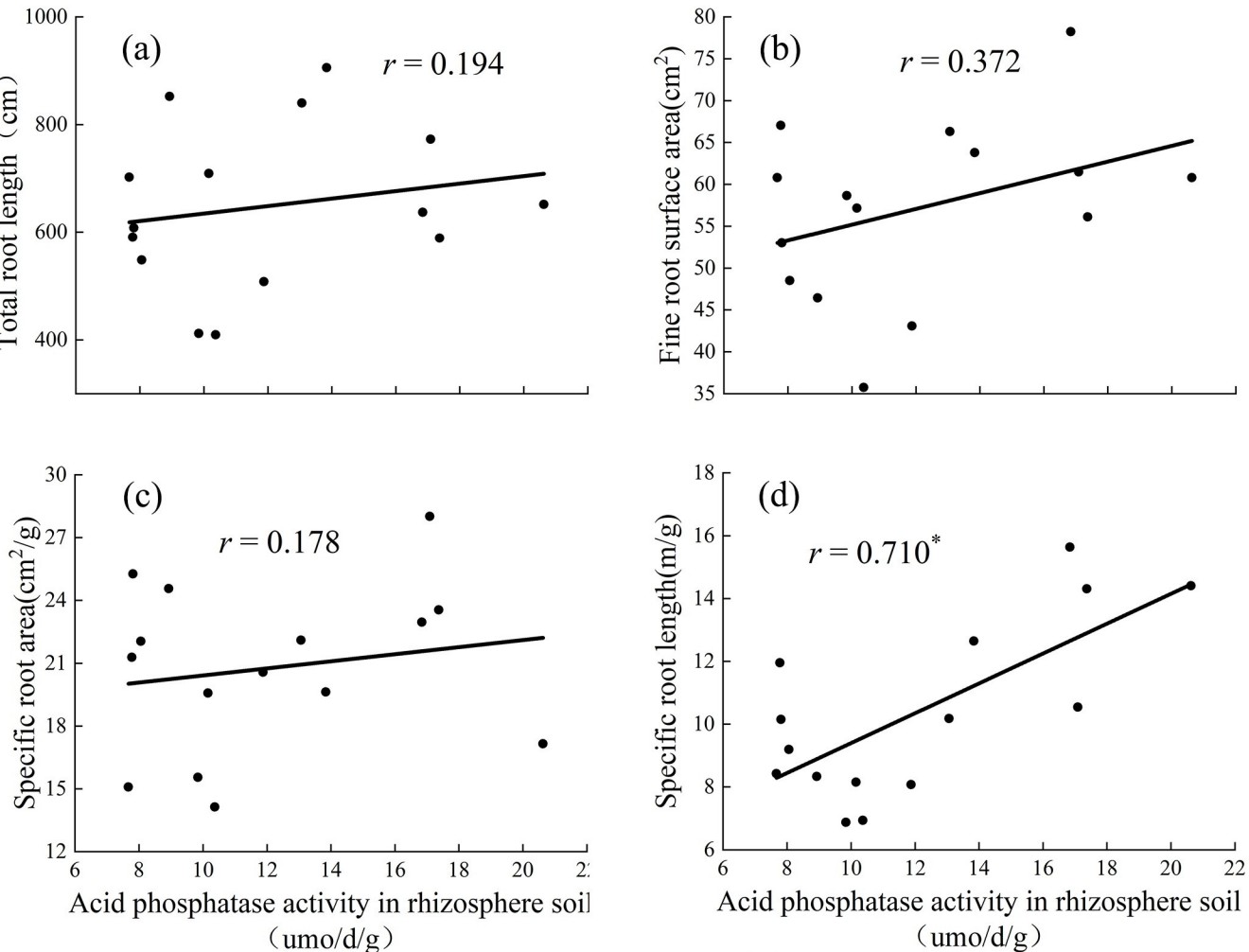

**Fig 7.** Relationships between the Acid phosphatase activity in rhizosphere soil and total root length (a), fine root surface area (b), specific root area (c) and specific root length (d) in loess soil. (The graph uses the statistical method of correlation analysis).

with flower quantity and leaf area and negatively correlated with plant height (Fig 6B and 6D and 6F.

### Relationship between acid phosphatase activity and root indices

When the substrate was loess, acid phosphatase activity in rhizosphere soil was strongly positively correlated with specific root length (Fig 7D), and had a positive correlation with total root length, fine root surface area, and specific root area (Fig 7A–7C).

When the substrate was vegetable soil, acid phosphatase activity in the rhizosphere soil had a significant negative correlation with fine root surface area (Fig 8B), a strong positive correlation with specific root area (Fig 8C), and a positive correlation with total root length and specific root length (Figs 8A and 8D, 9).

### Discussion

The indicators of plant growth and morphology include aboveground and underground parts. The aboveground parts are usually obvious and form the basis for observing plant growth.

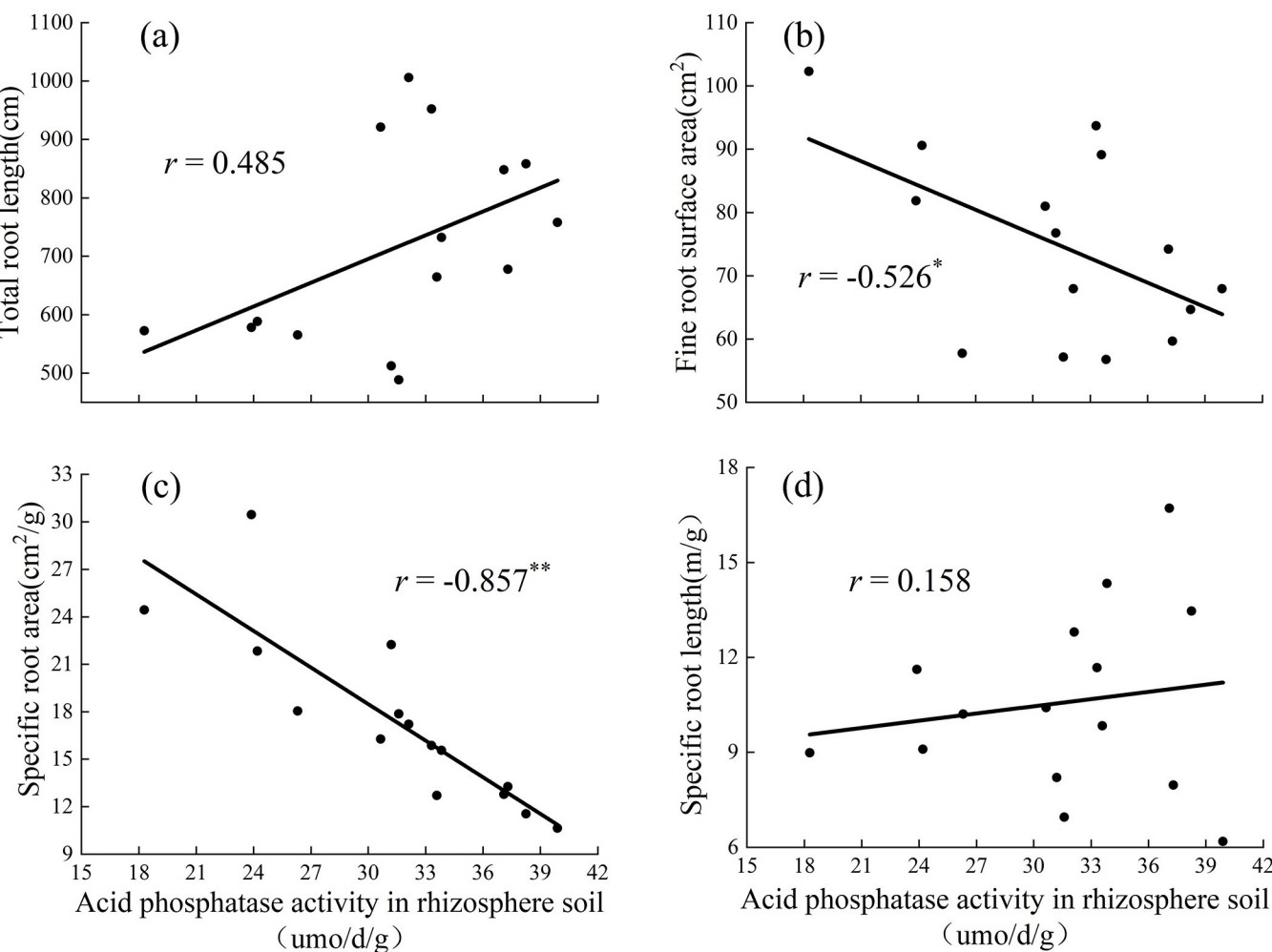

**Fig 8.** Relationships between the Acid phosphatase activity in rhizosphere soil and total root length (a), fine root surface area (b), specific root area (c) and specific root length (d) in vegetable soil. (The graph uses the statistical method of correlation analysis).

Underground parts reflect the adaptation of plants to soil with different nutrient concentrations. At different P levels, the growth indicators and morphology of *D. barbatus* changed to varying degrees. P is the third major nutrient required for plant growth and therefore plays an essential role [19]. Under P deficiency conditions, plant height, leaf area, and specific leaf area were smaller than in the other treatments (Fig 1). Furthermore, the root/shoot ratio in vegetable soil was significantly higher than that in other treatments (Fig 3), and the root/shoot ratio in loess soil was also higher than that in other treatments (Fig 3). This means that under P deficient conditions, the growth of aboveground parts will be limited. This will influence the distribution of plant nutrients to enhance the absorbency of the underground parts to improve root growth, which allows the roots to absorb more available P to accomplish plant growth. Under the low- and medium-to-low-P treatments, plant height, leaf area, specific leaf area, and leaf tissue density were significantly higher than those under the other treatments (Figs 1 and 2), indicating that low-P promotes the growth of the aboveground parts.

Specific leaf area can reflect the ability to obtain environmental resources. A lower specific leaf area can help plants cope better with poor environments while a higher specific leaf area can help plants to store nutrients [20–22]. In the present study, the specific leaf area in the two

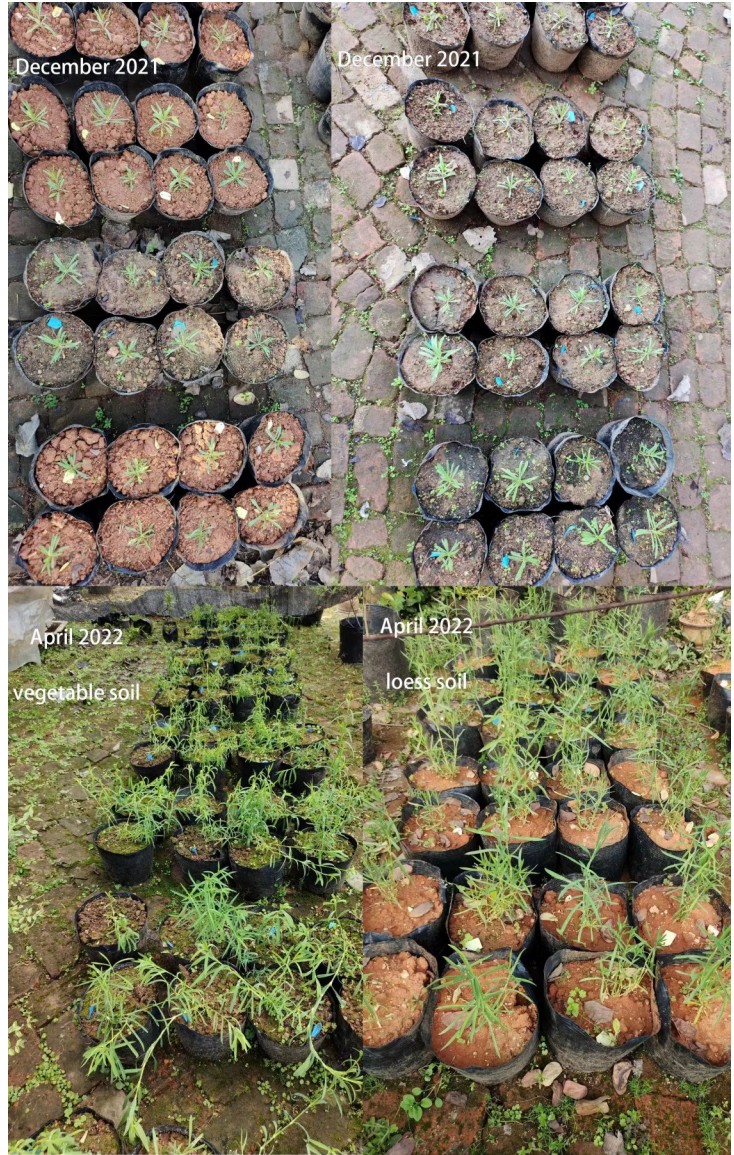

**Fig 9. *D. barbatus* growth photo.**

substrates was prominently increased in the low- and medium-P treatments (Fig 2). Compared with the control group, the addition of P can increase the specific leaf area of plants, indicating that moderate P conditions can improve the specific leaf area of *D. barbatus* and enhance the ability of its leaves to access environmental resources.

In a P-deficient environment, the structure of the cell membrane is destroyed, resulting in a decrease in the selective permeability of the cell membrane, which extravasates from the cell interior, compromising the cell's internal environment, physiological functions, and metabolism, and leading to an increase of blade relative conductivity [23]. Chloroplasts are organelles that participate in photosynthesis, and the chlorophyll content affects the growth rate of plants [24]. P is crucial for chloroplast production [25]. In the present study, the relative conductivity of *D. barbatus* leaves was not the same under the different P levels in the two substrates; the

relative conductivity of leaves in loess substrate was the highest under the $H_1$ treatment whereas in the vegetable soil it was the highest under the $C_4$ treatment (Fig 4). The cell membranes of plants were damaged to varying degrees in the two substrates. Under $H_1$ treatment in loess, the selective permeability of cell membranes was seriously damaged, plant damage was extensive, and the metabolic function of cells was disturbed, affecting the growth and development of plants. However, in the vegetable soil, this situation occurred under the $C_4$ treatment. Correlation analysis showed that the relative conductivity of loess soil blades was negatively correlated with the flower number, leaf area, and specific leaf area; that is, the increase in blade relative conductivity affects plant leaf area and inhibits reproduction (Fig 5). The relative conductivity of the vegetable soil leaves was positively correlated with plant height, flower number, leaf area, and specific leaf area (Fig 6). This may be because the acid phosphatase activity is generally high in vegetable soil to reduce damage to plants. Chlorophyll content reflects the photosynthetic ability of plant leaves. In the loess substrate, chlorophyll a and b contents increased under the $H_1$ and $H_2$ treatments (Table2). However, in vegetable soil, these increased under the $C_4$ treatment (Table2). This may be because the chlorophyll content is increased to alleviate the damage caused by P stress to plants when the blade relative conductivity is high. The increase in chlorophyll content improved the photosynthesis of *D. barbatus*. The number of flowers increased under the $H_1$ and $H_2$ treatments in loess substrate and under the $C_4$ treatment in vegetable soil (Fig 1), which has the same relationship with chlorophyll. The total chlorophyll content in the loess substrate was positively correlated with the number of flowers (Fig 5), and the total chlorophyll of the vegetable soil was positively correlated (Fig 6). This means that the $H_1$ and $H_2$ treatments of loess and the $C_4$ treatment of vegetable soil can improve the reproductive performance of *D. barbatus* and increase its number of flowers.

Organic P in soil cannot be directly absorbed by plants; it can only be absorbed and utilized by plants as inorganic P. The mineralization and decomposition of organophosphate depend on soil acid phosphatase activity, which converts organophosphorus into inorganic P in the soil [26, 27]. Under P deficient conditions, soil acid phosphatase activity was significantly enhanced, mainly because P deficiency induced an increase in the acid phosphatase activity of roots, thereby promoting the conversion of organophosphorus into inorganic P [28]. Some studies have shown that lower P concentrations can significantly increase root acid phosphatase activity [29], and the activity of acid phosphatase in rhizosphere soil was significantly increased under the low P treatment on both substrates (Fig 4). The vegetable soil was significantly affected by P deficiency, indicating that roots in environments with low P concentrations can stimulate microorganisms to secrete more acid phosphatase. Overall, the enhancement of phosphatase activity was a response to P deficiency in *D. barbatus*, which is consistent with the conclusion of Xu Jing's research [12]. The main cultivated soil in southern China is loess. Vegetable soils are usually obtained from the topsoil of vegetable gardens or leguminous crops, which have good fertility and agglomerate structure, while loess soil is slightly acidic and relatively poor in nutrients [3]. Maseko et al. [30] showed that the acid phosphatase activity of rhizosphere soil in legumes was significantly higher than that of non-rhizosphere soil. In the present study, the acid phosphatase activity was higher in vegetable soil than in loess for both bulk soil and rhizosphere soil. Therefore, in vegetable soil, *D. barbatus* can absorb more inorganic P to improve the number of flowers, and high P treatment can promote plant growth (Fig 1).

The morphology of the roots also changed with different P concentrations. Many studies have shown that under suitable P treatment, plants reduce the elongation of taproot cells to seek more P sources for growth and increase the lateral root length and root tissue density [31]. The root is the first to sense if the plant is under P deficiency stress and will change its many pattern indicators to adapt to P deficiency [32]. The total root length in the environment

of P deficiency was significantly shorter than that in an environment with adequate P [33, 34]. The larger the root volume, the more developed the root system of the plant, the larger the contact area between the root system and soil, and the stronger the ability of the plant to absorb nutrients [35, 36]. In the preset study, the increase in P concentration, total root length, root volume, etc. first increased and then decreased (Fig 3), and showed significant differences. The root system of *D. barbatus* was therefore sensitive to changes in soil P concentration and underwent significant changes, whereas treatment with no-P inhibited the growth of *D. barbatus* roots, and its total root length, root volume, and specific root length were all lower than those in the low- and medium-P treatments (Fig 3). This showed that lower P levels can promote the root growth of *D. barbatus* and its root/shoot ratio, allowing the plants to absorb enough available P through a developed root system to maintain the normal growth. Correlation analysis showed that the activity of acid phosphatase in rhizosphere soil was positively correlated with the total root length. The lower concentration of phosphorus application increased the activity of soil phosphatase, while the root length increased, and more phosphorus was absorbed into the surrounding soil (Figs 7 and 8). The $C_4$ treatment of the vegetable soil promoted the growth of aboveground parts and reduced the root/shoot ratio. This can help *D. barbatus* blossom more flowers. The $H_4$ treatment of the loess also reduced root growth and plants showed poor growth compared to that in vegetable soil. This may be because the activity of soil acid phosphatase is lower in loess than in vegetable soil.

## Conclusion

In conclusion, in the low-(0.192 g $Kg^{-1}$) and medium-to-low-P(0.384 g $Kg^{-1}$) treatments, the indicators of both aboveground and underground parts improved and promoted plant growth, and higher or lower than this range will not be conducive to the growth of *D. barbatus*. Compared with other treatments, the chlorophyll content, number of flowers, and activity of acid phosphatase increased in the rhizosphere soil of *D. barbatus* in the $H_1$ and $H_2$ treatments of loess and the $C_4$ treatment of vegetable soil. This shows that by adjusting its own regulatory system, *D. barbatus* had different physiological responses to different P treatments in the two substrates to adapt to changes in P concentration and ensure its normal growth.

## Supporting information

**S1 Table. Raw data for all charts.**
(XLSX)

## Acknowledgments

The author would like to thank Shaoyang University for providing the experimental sites. We would also like to thank the Shaoyang Forestry Bureau for its support for our technology and data.

## Author Contributions

**Data curation:** Xin Bin, Yike Qin, Shengping Zhao.

**Formal analysis:** Zhihuan Mao, Shengping Zhao.

**Investigation:** Ling Liu.

**Methodology:** Danhong Yin, Tengfei Huang.

**Project administration:** Aoqin Ma.

**Supervision:** Aoqin Ma.

**Writing – original draft:** Gui Chen.

**Writing – review & editing:** Gui Chen, Zhihuan Mao, Danhong Yin.

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
