## [Decision Letter · Decision Letter 0]

2 Aug 2023

PONE-D-23-20502Effects of different phosphorus levels on morphology and physiology of Dianthus barbatus LinnPLOS ONE

Dear Dr. Yin,

Thank you for submitting your manuscript to PLOS ONE. After careful consideration, we feel that it has merit but does not fully meet PLOS ONE’s publication criteria as it currently stands. Therefore, we invite you to submit a revised version of the manuscript that addresses the points raised during the review process.

We look forward to receiving your revised manuscript.

Kind regards,

Mojtaba Kordrostami, Ph.D.

Academic Editor

PLOS ONE

Journal Requirements:

The author would like to thank Natural Science Foundation of Hunan Province and National innovation (2022JJ50228) and Entrepreneurship training program for college students (202110547016) for providing financial support. 

Additional Editor Comments:

Dear Authors,

Thank you for submitting your manuscript to PLOS ONE. Your manuscript has been reviewed by four experts in the field. Based on their comments, we have decided to offer you the opportunity to revise your manuscript. Please find below the detailed comments from the reviewers.

Reviewer 1 raised concerns about the clarity and readability of the manuscript. They suggested that the manuscript needs to be rewritten before resubmission. They also pointed out that the "Introduction" lacks sufficient information explaining the focus of the study, and the "Materials and Methods" section lacks clarity on certain aspects. They also found inconsistencies in the content and redundancy in the description of measurement methods. The reviewer also noted that the "Results" section contains repetition and confusing statements, and the image quality is poor. They recommended using the Pearson correlation coefficient or other analysis methods to present and describe the correlation analysis of different indicators. In the "Discussion" section, the reviewer suggested indicating which figure or table the discussion of the relevant content comes from.

Reviewer 2 suggested that the expression "middle, high, no-P" is not clear and recommended using different processing groups such as H0, H1, H2, H3, H4, etc. They also pointed out that ANOVA should be performed before concluding. They noted that capital letters indicating differences between substrates are not seen in Figure 1 and asked why the chlorophyll content was not differently analyzed according to the matrix. They also recommended modifying the title because the content involves different soil substrates and providing detailed ingredient lists for the two different substrates.

Reviewer 3 found the research interesting but suggested adding the significance of the plant to the local soils and communities to attract a range of readers. They also pointed out several areas in the manuscript that need correction or improvement, such as the use of English, the use of terms, and the quality of figures. They also suggested adding p values and linear equations to the figures to help readers study the relationships.

Reviewer 4 found the research potentially helpful for the agricultural and industrial fields but raised concerns about the data representation and the quality of the figure images. They also suggested that the discussion can be significantly improved by citing previous literature. They provided a list of specific questions and comments for the paper, which you should address in your revision.

In light of these comments, we invite you to revise your manuscript. Please ensure that you address all the points raised by the reviewers in your revised manuscript and in your response to reviewers.

We look forward to receiving your revised manuscript.

Best regards,

Mojtaba Kordrostami

Editor, PLOS ONE

Reviewers' comments:

Reviewer's Responses to Questions

**Comments to the Author**

1. Is the manuscript technically sound, and do the data support the conclusions?

Reviewer #1: No

Reviewer #2: Yes

Reviewer #3: Yes

Reviewer #4: No

2. Has the statistical analysis been performed appropriately and rigorously? 

Reviewer #1: Yes

Reviewer #2: Yes

Reviewer #3: I Don't Know

Reviewer #4: No

3. Have the authors made all data underlying the findings in their manuscript fully available?

Reviewer #1: Yes

Reviewer #2: Yes

Reviewer #3: No

Reviewer #4: Yes

4. Is the manuscript presented in an intelligible fashion and written in standard English?

Reviewer #1: No

Reviewer #2: Yes

Reviewer #3: Yes

Reviewer #4: Yes

5. Review Comments to the Author

Reviewer #1: The paper used Dianthus as the experimental material to analyze the changes in its morphology and physiological indicators by varying the phosphorus content in two different substrates. Overall, the experimental design of the paper is satisfactory, and the research methods used are conventional. There are many writing problems in the manuscript, and there were redundancies or insufficient explanations in some parts. The manuscript does not yield sufficient valuable results, and is almost chaotic and not readable. The manuscript needs to be rewritten before resubmission, therefore it is recommended to reject it.

Specifically,

In “Introduction”, sufficient information is not provided to explain why the study focused on the phenotypic and physiological research of Dianthus under different phosphorus levels. Additionally, there is a lack of relevant reports on the relationship between photosynthesis, the tested indicators, and phosphorus.

In the “Materials and Methods” section, it is unclear whether the initial phosphorus content of the loess and vegetable soil used was determined. Why were these two substrates chosen? What was the basis for selecting the five phosphorus levels for treatment? What was the amount of phosphorus used in the control treatment? There are inconsistencies in the content, for example, line 76 mentions that "the morphological and physiological characteristics of its root system were analyzed," but subsequent sections measure indicators that include above-ground parts. In Table 1, what do H and C represent? This needs to be explained. What is low P and what is high P? The section title should be "Materials and Methods" instead of "Materials and research methods". The description of the measurement methods for morphological indicators is too redundant, such as Root biomass and Leaf biomass, etc. Line 101, what does "5L" mean? What does "3.5 mL of the solution" refer to? The description of the soil acid phosphatase activity measurement method is too simple. What is the calculation formula for this enzyme? Line 129 includes two A633.

In “Results”, there is repetition in the following two sentences: "The height of D. barbatus in the low-P treatment was the highest, which was significantly higher than that in the other treatments" and "The effects of middle- and low-P treatments on plant height were slight but significantly higher than that of medium-, high-, and no-P treatments." The meaning of the sentence "The flower quantity in the low-, middle-, and low-P treatments was significantly higher than that in the medium-, high-, and no-P treatments" is confusing. Where are the capital letters in Fig. 1 which indicate significant differences between two substrates with the same phosphorus treatment at the 0.05 level? In the "Leaf/Root morphological index" section, the statements are also confusing, repetitive, and lack emphasis. The image quality is poor and not clear. It is recommended to use the Pearson correlation coefficient or other analysis methods and software to present and describe the correlation analysis of different indicators.

In the “Discussion” section, it is necessary to indicate which figure or table the discussion of the relevant content comes from. The reason for choosing two different soil substrates was not discussed. Specific meaningful conclusions were not obtained.

Reviewer #2: 1."middle, high, no-P" expression is not clear, it is suggested that the full text of middle, high, no-P into the picture of different processing groups such as H0, H1, H2, H3, H4, etc.

2.Line151：“Under the different levels ofP supply, D. barbatus plants in vegetable soil were generally taller and had more flower （Fig.1）. Thus, vegetable soil was helpful for the vegetative and reproductive growth of this species.”ANOVA should be performed and then concluded.

3.Line 156“Different capital letters indicate significant differences between two substrates with the same phosphorus treatment at the 0.05 level. ”Capital letters indicating differences between substrates are not seen in Figure 1.

4.Why is the chlorophyll content not differently analyzed according to the matrix (Table 2)?

5.Line 89：“48h”

6.The conclusion part needs to be perfected.

7.It is recommended to modify the title because the content involves different soil substrates.

8.Please provide detailed ingredient lists for the two different substrates. The manuscript does not go into detail about what both substrates contain.

Reviewer #3: General comments

Studying the morphological and physiological responses of Dianthus barbatus grown on loess and vegetable soil treated with varying P levels, may help understand plant adaptive strategies to varying growth conditions. However, in the manuscript, I could not find significance of this plant to the local soils and communities, and this must be added to attract a range of readers/interest. This is a simple research described with clarity and may be considered for publication once all readers comments are satisfied.

Specific comments

line 14: ratio is a relative term while the very first sentence uses this term as an absolute; needs correction. Still, this sentence makes incorrect use of English; must be reworded and improved. what does "its" refers to here?

line 17: line may better start with "Two substrates". Better use "belowground" instead of "underground".

line 21: Clarify what means flower quantity - is it number of flowers or size or what exactly.

line 25-26: I like the take-home message at the end and it would increase the interest of readers. As well, a single sentence at the start of the abstract signifying why the plant was studied would give initiative to reader to dig deep or read further.

line 28: using word "main" is ambiguous - why don't authors use "one of the major nutrient elements"?

lines 34-37: authors must take a note that loess is different than most agricultural or forest soils; therefore, they should let the readers know every now and then (where helpful) through the paper that the study was conducted in a loess/soil. for example, lines 34-37 and objective lines 62-64 must mention "loess".

Intro: it would be much better to add some significance of loess and the study plant to make the intro interesting.

line 67: provide coordinates to help readers know exact location and the climate. Also, add a figure showing the region and illustrating the experimental design or treatment plan.

line 107: what was the temperature?

line 112: authors may want to use plural

figures: significance letters on top of bars are so small fonts that the use of these letters becomes useless. Also, the r values, and the axes titles are not readable unless zoomed in 2x or more. The figures must be readable to help readers and maintain the quality requirements of this PLOS.

figures: add p values and linear equation to help readers study the relationships.

line 427: use "sustain" in place of "ensure"

Reviewer #4: Phosphorus limitation largely impacts plant growth. The authors have investigated the above- and under-ground traits of Dianthus barbatus, an agriculture important flower in China, under the various phosphorus treatments using two substrates, loess and vegetable soil. The authors described how the different phosphorus treatments impacts root or leaf growth by measuring several plant parameters. The results in this paper might be helpful for the agricultural and industrial fields to reduce P fertilization because D. barbatus is one of the important flowers in China and loess are indicated as common materials to cultivate and landscape in China. However, although English is largely fine, the data representation is still needed to be largely improved. It is hard to judge whether what the authors have claimed as significant difference is really the case. The figure image quality should be improved significantly. The discussion can be significantly improved by citing the previous literatures.

I have the following questions and comments for this paper.

1. Abstract: Thus, D. barbatus seems to reduce the damage caused by phosphorus stress by increasing chlorophyll content and root acid phosphatase activity. → Thus, D. barbatus seems to reduce the growth retardation caused by phosphorus stress by increasing chlorophyll content and root acid phosphatase activity. I am not also quite sure whether the plant increases chlorophyll content to reduce the growth retardation. Do you have enough evidence that connects increased chlorophyll content and plant adaptation to low Pi?

2. Since the majority of people do not know about the plant, to better understand the plant, it would be necessary to show the growth under different phosphorus conditions by images. The quantitative data only is not enough for this.

3. Line 47,The content of organic P in the rhizosphere soil was significantly higher than that 48 in the non-rhizosphere soil [10]. → I am not quite sure about this statement. The content of organic P should be reduced due to the activity of acid phosphatase??

4. In lines 71-72, Table 1, and Figure 1, it is indicated that two substrates and five phosphorus treatments are used in this paper. However, it is difficult to understand the differences between the Hn and Cn treatments because you haven’t shown any relevant details in the beginning. I recommend specifying the differences in the text (around lines 71-72) and the caption of Figure 1. The information is very critical for this study.

5. In lines 89-90, does “for 48” mean “for 48 h”?

6. In lines 148-149, you should change the “D” in “D. barbatus” to italic.

7. In the text of the results, you used the no-P, low-P, middle- and low-P, medium-P, medium- and low-P, medium-to-low-P, and high-P treatments. I suggest you specify each treatment (Hn and Cn) correspondence to each phosphorus treatment the first time. (e.g., low-P (H1 and C1)) Also, I recommend unifying the wording.

8. In the Figures, image resolution is not enough to read the x-axis, y-axis, and legend words on my environment. I recommend changing the font sizes and figure format. 28. 48 in the non-rhizosphere soil [10].

9. Related to 7, I feel that the description of the legends is not enough. For example, I don't get that what kinds of statistical methods were used in each figure. This is especially related to Fig.2, Fig.3, Fig.4, Fig.5, Fig.6, Fig.7, Fig.8.

10. Related to point 8, In lines 151-153 and 156-157, you present, “D. barbatus plants in vegetable soil were generally taller”. However, I cannot agree with this remark because the plant height in loess and vegetable soil are not significant differences in Figure 1. Have you forgotten to show the results of the statistical analysis? If you have the data, it is better to write.

11. In lines 163-165 and Figure 2b, you present, “The specific leaf area was largest under the middle- and low-P treatments, which was significantly higher than that under the no-, low-, and medium-P treatments (Fig.2b)”. However, the specific leaf area of H1 treatment looks similar level as the H4 treatment.

12. In lines 166-168 and Figure 2c, I cannot agree with your opinion because the no-P (Hc treatment) data does not show the result of statistical analysis.

13. In lines 168-170 and Figure 2d, the H1 data shows the same trends as the H3 treatment, according to the statistical analysis.

14. In lines 210-213, two sentences have the same statement.

15. In line 218, I think it is a mistake. Isn’t the correct information “When the substrate was vegetable soil”?

16. In lines 222-223, I think it is a mistake. Isn’t the correct information “The specific root length reached the maximum in the middle- and low-P treatment”?

17. In line 244, “root tissue density (f)” should be instead described as “specific root area (f)”.

18. In the caption of Figure 4, you should change the “Dianthus barbatus” to italic. In addition, only this caption is attached as the image.

19. In the discussion, I recommend citing the Figure number when you want to mention the results.

20. You suggest improving root growth and absorbency for the P acquisition under low-P conditions. In the data, are root growth indicators improved under no-P conditions? If plants ensure normal growth under P-limited conditions, root elongation seems to be promoted under no-P conditions.

6. PLOS authors have the option to publish the peer review history of their article (what does this mean?). If published, this will include your full peer review and any attached files.

Reviewer #1: No

Reviewer #2: No

Reviewer #3: No

Reviewer #4: No

---

## [Author Response · Author response to Decision Letter 0]

21 Sep 2023

September 14, 2023

Dr. Mojtaba Kordrostami Editor, PLOS ONE

Re: Response for manuscript “Effects of two substrates at different phosphorus levels on morphology and physiology of Dianthus barbatus Linn”

Dear Dr. Mojtaba Kordrostami Editor, PLOS ONE,

Thanks for providing us with this great opportunity to submit a revised version of our manuscript. We appreciate the detailed and constructive comments provided by the reviewers. We have carefully revised the manuscript by incorporating all the suggestions by the review panel.

We hope this revised manuscript has addressed your concerns, and look forward to hearing from you.

Sincerely,

The Authors

Encl. Responses to the comments from Reviewer 1, 2, 3, 4.

Reply to Reviewer #1

Dear Reviewers,

Thank you very much for your time involved in reviewing the manuscript and your very encouraging comments on the merits.

Comments: 

“The paper used Dianthus as the experimental material to analyze the changes in its morphology and physiological indicators by varying the phosphorus content in two different substrates. Overall, the experimental design of the paper is satisfactory, and the research methods used are conventional. There are many writing problems in the manuscript, and there were redundancies or insufficient explanations in some parts. The manuscript does not yield sufficient valuable results, and is almost chaotic and not readable. The manuscript needs to be rewritten before resubmission, therefore it is recommended to reject it.”

We also appreciate your clear and detailed feedback and hope that the explanation has fully addressed all of your concerns. In the remainder of this letter, we discuss each of your comments individually along with our corresponding responses. 

To facilitate this discussion, we first retype your comments in italic font and then present our responses to the comments. 

Comment 1:

In “Introduction”, sufficient information is not provided to explain why the study focused on the phenotypic and physiological research of Dianthus under different phosphorus levels. Additionally, there is a lack of relevant reports on the relationship between photosynthesis, the tested indicators, and phosphorus. 

Response 1:

Thank you for the detailed review. In “Introduction”, We emphasize that it is of great significance to understand the physiological mechanism of phosphorus uptake and utilization in plants and the physiological response of plants to phosphorus deficiency stress, to find the most suitable phosphorus concentration for plant growth, and to improve the efficiency of phosphorus uptake and utilization in plants.There are many studies on the effects of P addition on plant roots worldwide but there has been no research on how P affects the roots of D. barbatus.Therefore,we are study focused on the phenotypic and physiological research of Dianthus under different phosphorus levels.

Comment 2:

In the “Materials and Methods” section, it is unclear whether the initial phosphorus content of the loess and vegetable soil used was determined. Why were these two substrates chosen? What was the basis for selecting the five phosphorus levels for treatment? What was the amount of phosphorus used in the control treatment? There are inconsistencies in the content, for example, line 76 mentions that "the morphological and physiological characteristics of its root system were analyzed," but subsequent sections measure indicators that include above-ground parts. In Table 1, what do H and C represent? This needs to be explained. What is low P and what is high P? The section title should be "Materials and Methods" instead of "Materials and research methods". The description of the measurement methods for morphological indicators is too redundant, such as Root biomass and Leaf biomass, etc. Line 101, what does "5L" mean? What does "3.5 mL of the solution" refer to? The description of the soil acid phosphatase activity measurement method is too simple. What is the calculation formula for this enzyme? Line 129 includes two A633. 

Response 2:

Thanks for your great suggestion on improving the accessibility of our manuscript. 1.In the “Materials and Methods” section, We label phosphorus content of the loess and vegetable soil. 2.The main cultivated soil in southern China is loess , and vegetable soil is slightly alkaline, and the soil nutrients are relatively fertile. 3.We referred to other literature on phosphorus treatment to specify a phosphorus gradient specifically for D. barbatus. 4. The amount of phosphorus in the control treatment was zero. 5.The "form" here represents the above-ground parts and above-below parts, we are very sorry for the perplex. 6.The "H" here represents the loess and "C" here represents thevegetable soil. 7.We have changed to “Materials and Methods”. 8.We have tried to simplify the presentation of the experimental method. 9.We have fixed the error in line 101. 10.The "3.5 mL" here represents the Oscillating liquid. 11.We added soil acid phosphatase activity measurement method and calculation formula for this enzyme. 12.We have corrected the error.

Comment 3:

In “Results”, there is repetition in the following two sentences: "The height of D. barbatus in the low-P treatment was the highest, which was significantly higher than that in the other treatments" and "The effects of middle- and low-P treatments on plant height were slight but significantly higher than that of medium-, high-, and no-P treatments." The meaning of the sentence "The flower quantity in the low-, middle-, and low-P treatments was significantly higher than that in the medium-, high-, and no-P treatments" is confusing. Where are the capital letters in Fig. 1 which indicate significant differences between two substrates with the same phosphorus treatment at the 0.05 level?

Response 3:

Thanks for your great suggestion on improving the accessibility of our manuscript. 1.We have corrected the error. 2.Thank you for the detailed review.We have moved this sentence to Figure 2 below.

Comment 4:

In the "Leaf/Root morphological index" section, the statements are also confusing, repetitive, and lack emphasis. The image quality is poor and not clear. It is recommended to use the Pearson correlation coefficient or other analysis methods and software to present and describe the correlation analysis of different indicators.

Response 4:

Thanks for your great suggestion on improving the accessibility of our manuscript.We are very sorry for the trouble caused to you. We have changed the clear picture.We used Excel2019 for the original data processing in the early stage, and related statistical analysis such as one-way ANOVA analysis, normality test, independent sample T test.

Comment 5:

In the “Discussion” section, it is necessary to indicate which figure or table the discussion of the relevant content comes from. The reason for choosing two different soil substrates was not discussed. Specific meaningful conclusions were not obtained.

Response 5:

Thanks for your great suggestion on improving the accessibility of our manuscript.We are indicate which figure or table the discussion of the relevant content comes from.We briefly explain the reasons for choosing two different soil substrates.

We would like to take this opportunity to thank you for all your time involved and this great opportunity for us to improve the manuscript. We hope you will find this revised version satisfactory. 

Sincerely,

The Authors

Reply to Reviewer #2

Dear Reviewers,

Thank you very much for your time involved in reviewing the manuscript and your very encouraging comments on the merits.We also appreciate your clear and detailed feedback and hope that the explanation has fully addressed all of your concerns. In the remainder of this letter, we discuss each of your comments individually along with our corresponding responses. 

To facilitate this discussion, we first retype your comments in italic font and then present our responses to the comments.

Comment 1:

 suggested that the expression "middle, high, no-P" is not clear and recommended using different processing groups such as H0, H1, H2, H3, H4, etc.

Response 1:

Thank you for the detailed review. We have carefully and thoroughly proofread the manuscript to replace no-P , low, low-middle, middle, high with H0(C0), H1(C1), H2(C2), H3(C3), H4(C4).

Comment 2:

Line151：“Under the different levels ofP supply, D. barbatus plants in vegetable soil were generally taller and had more flower （Fig.1）. Thus, vegetable soil was helpful for the vegetative and reproductive growth of this species.”ANOVA should be performed and then concluded.

Response 2:

Thank you for the detailed review.All of our conclusions are based on one-way ANOVA analysis of variance

Comment 3:

Line 156“Different capital letters indicate significant differences between two substrates with the same phosphorus treatment at the 0.05 level. ”Capital letters indicating differences between substrates are not seen in Figure 1.

Response 3:

Thank you for the detailed review.We have moved this sentence to Figure 2 below.

Comment 4:

Why is the chlorophyll content not differently analyzed according to the matrix (Table 2)?

Response 4:

We focused more on morphological analysis, so we did not analyze chlorophyll content differently depending on the substrate.

Comment 5:

Line 89：“48h”

Response 5:

Thank you for the detailed review.The new manuscript has been revised

Comment 6:

The conclusion part needs to be perfected

Response 6:

Thank you for the detailed review.We have made some changes to the conclusion.

Comment 7:

It is recommended to modify the title because the content involves different soil substrates.

Response 7:

Thank you for the detailed review.The new title has been revised

Comment 8:

Please provide detailed ingredient lists for the two different substrates. The manuscript does not go into detail about what both substrates contain.

Response 8:

Thank you for the detailed review.We have added a new ingredient list at the end of the article.

We would like to take this opportunity to thank you for all your time involved and this great opportunity for us to improve the manuscript. We hope you will find this revised version satisfactory. 

Sincerely,

The Authors

Reply to Reviewer #3

Dear Reviewers,

Thank you very much for your time involved in reviewing the manuscript and your very encouraging comments on the merits.

Comments: 

“General comments

Studying the morphological and physiological responses of Dianthus barbatus grown on loess and vegetable soil treated with varying P levels, may help understand plant adaptive strategies to varying growth conditions. However, in the manuscript, I could not find significance of this plant to the local soils and communities, and this must be added to attract a range of readers/interest. This is a simple research described with clarity and may be considered for publication once all readers comments are satisfied.”

We also appreciate your clear and detailed feedback and hope that the explanation has fully addressed all of your concerns. In the remainder of this letter, we discuss each of your comments individually along with our corresponding responses. 

To facilitate this discussion, we first retype your comments in italic font and then present our responses to the comments. 

Comment 1:

line 14: ratio is a relative term while the very first sentence uses this term as an absolute; needs correction. Still, this sentence makes incorrect use of English; must be reworded and improved. what does "its" refers to here?

Response 1:

Thank you for the detailed review.We apologize for the inconvenience, and we have taken care to correct the situation.The term "its" represents the Dianthus barbatus.

Comment 2:

line 17:line may better start with "Two substrates". Better use "belowground" instead of "underground".

Response 2:

Thank you for the detailed review.We have corrected the error.

Comment 3:

line 21: Clarify what means flower quantity - is it number of flowers or size or what exactly.

Response 3:

Thank you for the detailed review.we are sorry for the trouble. This refers to the number of flowers

Comment 4:

line 25-26:   I like the take-home message at the end and it would increase the interest of readers. As well, a single sentence at the start of the abstract signifying why the plant was studied would give initiative to reader to dig deep or read further.

Response 4:

Thank you for the detailed review.We have increased the importance of studying D. barbatus according to what you said.

Comment 5:

line 28: using word "main" is ambiguous - why don't authors use "one of the major nutrient elements"?

Response 5:

Thank you for the detailed review.we are sorry for the trouble. We have corrected the error

Comment 6:

lines 34-37: authors must take a note that loess is different than most agricultural or forest soils; therefore, they should let the readers know every now and then (where helpful) through the paper that the study was conducted in a loess/soil. for example, lines 34-37 and objective lines 62-64 must mention "loess".

Intro: it would be much better to add some significance of loess and the study plant to make the intro interesting.

Response 6:

Thank you for the detailed review.We have added a new ingredient list at the end of the article.

Comment 7:

line 67: provide coordinates to help readers know exact location and the climate. Also, add a figure showing the region and illustrating the experimental design or treatment plan.

Response 7:

Thank you for the detailed review. We have carefully and thoroughly proofread the manuscript to replace no-P, low, low-middle, middle，high with H0(C0), H1(C1), H2(C2), H3(C3), H4(C4).

Comment 8:

line 107: what was the temperature?

Response 8:

Thank you for the detailed review.Sorry we forgot to add the temperature, we have corrected it.

Comment 9:

line 112: authors may want to use plural

figures: significance letters on top of bars are so small fonts that the use of these letters becomes useless. Also, the r values, and the axes titles are not readable unless zoomed in 2x or more. The figures must be readable to help readers and maintain the quality requirements of this PLOS.

figures: add p values and linear equation to help readers study the relationships.

Response 9:

Thank you for the detailed review.We have added a new ingredient list at the end of the article.

Comment 9:

line 427: use "sustain" in place of "ensure"

Response 9:

Thank you for the detailed review.We have added a new ingredient list at the end of the article.

We would like to take this opportunity to thank you for all your time involved and this great opportunity for us to improve the manuscript. We hope you will find this revised version satisfactory. 

Sincerely,

The Authors

Reply to Reviewer #4

Dear Reviewers,

Thank you very much for your time involved in reviewing the manuscript and your very encouraging comments on the merits.

Comments: 

“Phosphorus limitation largely impacts plant growth. The authors have investigated the above- and under-ground traits of Dianthus barbatus, an agriculture important flower in China, under the various phosphorus treatments using two substrates, loess and vegetable soil. The authors described how the different phosphorus treatments impacts root or leaf growth by measuring several plant parameters. The results in this paper might be helpful for the agricultural and industrial fields to reduce P fertilization because D. barbatus is one of the important flowers in China and loess are indicated as common materials to cultivate and landscape in China. However, although English is largely fine, the data representation is still needed to be largely improved. It is hard to judge whether what the authors have claimed as significant difference is really the case. The figure image quality should be improved significantly. The discussion can be significantly improved by citing the previous literatures.

I have the following questions and comments for this paper.”

We also appreciate your clear and detailed feedback and hope that the explanation has fully addressed all of your concerns. In the remainder of this letter, we discuss each of your comments individ

---

## [Decision Letter · Decision Letter 1]

29 Nov 2023

PONE-D-23-20502R1Effects of two substrates at different phosphorus levels on morphology and physiology of Dianthus barbatus LinnPLOS ONE

Dear Dr. Yin,

Thank you for submitting your manuscript to PLOS ONE. After careful consideration, we feel that it has merit but does not fully meet PLOS ONE’s publication criteria as it currently stands. Therefore, we invite you to submit a revised version of the manuscript that addresses the points raised during the review process.

We look forward to receiving your revised manuscript.

Kind regards,

Mojtaba Kordrostami, Ph.D.

Academic Editor

PLOS ONE

Journal Requirements:

Additional Editor Comments :

Please perform the minor revisions and send it back for final acceptance.

Reviewers' comments:

Reviewer's Responses to Questions

**Comments to the Author**

1. If the authors have adequately addressed your comments raised in a previous round of review and you feel that this manuscript is now acceptable for publication, you may indicate that here to bypass the “Comments to the Author” section, enter your conflict of interest statement in the “Confidential to Editor” section, and submit your "Accept" recommendation.

Reviewer #1: (No Response)

Reviewer #2: All comments have been addressed

2. Is the manuscript technically sound, and do the data support the conclusions?

Reviewer #1: (No Response)

Reviewer #2: Yes

3. Has the statistical analysis been performed appropriately and rigorously? 

Reviewer #1: (No Response)

Reviewer #2: Yes

4. Have the authors made all data underlying the findings in their manuscript fully available?

Reviewer #1: (No Response)

Reviewer #2: Yes

5. Is the manuscript presented in an intelligible fashion and written in standard English?

Reviewer #1: (No Response)

Reviewer #2: Yes

6. Review Comments to the Author

Reviewer #1: In this manuscript, what is the normal level of phosphorus supply for Dianthus? In the experiment, are the different phosphorus concentrations used considered high or low levels for Dianthus?

Please use one paragraph to introduce the main methods, kits, and calculation formulas in the "Soil acid phase activity" section. Do not describe trivial content such as "preparation" and "procedure", and etc.

What dose Fig. 9 mean?

Additionally, the manuscript should also pay attention to writing standards.

Reviewer #2: (No Response)

7. PLOS authors have the option to publish the peer review history of their article (what does this mean?). If published, this will include your full peer review and any attached files.

Reviewer #1: No

Reviewer #2: No

---

## [Author Response · Author response to Decision Letter 1]

7 Dec 2023

Encl. Responses to the comments from Reviewer 1, 2.

Reply to Reviewer #1

Dear Reviewers,

Thank you very much for your time involved in reviewing the manuscript and your very encouraging comments on the merits. We also appreciate your clear and detailed feedback and hope that the explanation has fully addressed all of your concerns. In the remainder of this letter, we discuss each of your comments individually along with our corresponding responses. 

To facilitate this discussion, we first retype your comments in italic font and then present our responses to the comments.

Comments 1: 

“In this manuscript, what is the normal level of phosphorus supply for Dianthus?”

Response 1:

Thank you for your detailed review. In the "Introduction", we emphasize finding the most suitable phosphorus concentration for D. barbatus growth and improving the absorption and utilization efficiency of phosphorus by D. barbatus. In "Materials and Methods", we mentioned that the blank treatment of this experiment used untreated soil. The loess soil has the following components: soil total phosphorus was 0.47 g/kg. The vegetable soil has the following components: soil total phosphorus was 0.85 g/kg.

Comment 2:

In the experiment, are the different phosphorus concentrations used considered high or low levels for Dianthus?

Response 2:

Thanks for your great suggestion on improving the accessibility of our manuscript. We are very sorry for the trouble caused to you. Under low-(0.192 〖g∙Kg〗^(-1)) and medium-to-low-P(0.384 〖g∙Kg〗^(-1)) treatments treatments, the indicators of both aboveground and underground parts improved and promoted plant growth, and higher or lower than this range will not be conducive to the growth of D. barbatus.

Comment 3:

Please use one paragraph to introduce the main methods, kits, and calculation formulas in the "Soil acid phase activity" section. Do not describe trivial content such as" preparation" and "procedure", and etc.

Response 3:

Thanks for your great suggestion on improving the accessibility of our manuscript. We are very sorry for the trouble caused to you. We have deleted the words "preparation" and "procedure" in the section "Soil acid phase Activity". And retain the parts that make the essay coherent.

Comment 4:

What dose Fig.9 mean?

Response 4:

Thanks for your great suggestion on improving the accessibility of our manuscript. We are very sorry for the trouble caused to you. The purpose of Figure 9 is to enhance readers' comprehension of D. barbatus by presenting photographs depicting their growth under varying phosphorus conditions. These images were captured in October 2021 and April 2022. The darker color is vegetable soil, and the other is loess.

Comment 5:

Additionally, the manuscript should also pay attention to writing standards.

Response 5:

Thanks for your great suggestion on improving the accessibility of our manuscript. We have modified the article according to the style and formatting guidelines.

We would like to take this opportunity to thank you for all your time involved and this great opportunity for us to improve the manuscript. We hope you will find this revised version satisfactory. 

Sincerely,

The Authors

Reply to Reviewer #2

Dear Reviewers,

Thank you very much for your time involved in reviewing the manuscript and your very encouraging comments on the merits and this great opportunity for us to improve the manuscript. We hope you will find this revised version satisfactory. 

Sincerely,

The Authors

---

## [Decision Letter · Decision Letter 2]

2 Jan 2024

Effects of two substrates at different phosphorus levels on morphology and physiology of Dianthus barbatus Linn.

PONE-D-23-20502R2

Dear Dr. Yin,

We’re pleased to inform you that your manuscript has been judged scientifically suitable for publication and will be formally accepted for publication once it meets all outstanding technical requirements.

Kind regards,

Mojtaba Kordrostami, Ph.D.

Academic Editor

PLOS ONE

Additional Editor Comments (optional):

The manuscript can be accepted now.

Reviewers' comments:

Reviewer's Responses to Questions

**Comments to the Author**

1. If the authors have adequately addressed your comments raised in a previous round of review and you feel that this manuscript is now acceptable for publication, you may indicate that here to bypass the “Comments to the Author” section, enter your conflict of interest statement in the “Confidential to Editor” section, and submit your "Accept" recommendation.

Reviewer #1: All comments have been addressed

Reviewer #2: All comments have been addressed

2. Is the manuscript technically sound, and do the data support the conclusions?

Reviewer #1: Yes

Reviewer #2: Yes

3. Has the statistical analysis been performed appropriately and rigorously? 

Reviewer #1: Yes

Reviewer #2: Yes

4. Have the authors made all data underlying the findings in their manuscript fully available?

Reviewer #1: Yes

Reviewer #2: Yes

5. Is the manuscript presented in an intelligible fashion and written in standard English?

Reviewer #1: Yes

Reviewer #2: Yes

6. Review Comments to the Author

Reviewer #1: (No Response)

Reviewer #2: (No Response)

7. PLOS authors have the option to publish the peer review history of their article (what does this mean?). If published, this will include your full peer review and any attached files.

Reviewer #1: No

Reviewer #2: No

---

## [Editor Report · Acceptance letter]

23 May 2024

PONE-D-23-20502R2 

PLOS ONE

Dear Dr. Yin, 

I'm pleased to inform you that your manuscript has been deemed suitable for publication in PLOS ONE. Congratulations! Your manuscript is now being handed over to our production team.

Kind regards, 

on behalf of

Dr. Mojtaba Kordrostami 

Academic Editor

PLOS ONE